# Comparative Assessment of Spire and COSMIC-2 Radio Occultation Data Quality

Cong Qiu [1,2], Xiaoming Wang [1,3,*], Kai Zhou [1], Jinglei Zhang [1], Yufei Chen [1,2], Haobo Li [4], Dingyi Liu [1,2] and Hong Yuan [1]

1 Aerospace Information Research Institute, Chinese Academy of Sciences, No.9 Dengzhuang South Road, Haidian District, Beijing 100094, China; qiucong@aircas.ac.cn (C.Q.); zhoukai@aircas.ac.cn (K.Z.); zhangjinglei@aircas.ac.cn (J.Z.); chenyufei20@mails.ucas.ac.cn (Y.C.); liudingyi21@mails.ucas.ac.cn (D.L.); yuanhong@aircas.ac.cn (H.Y.)
2 School of Electronic, Electrical and Communicating Engineering, University of Chinese Academy of Sciences, No. 19A Yuquan Road, Shijingshan District, Beijing 100049, China
3 College of Resources and Environment, University of Chinese Academy of Sciences, No. 19A Yuquan Road, Shijingshan District, Beijing 100049, China
4 School of Science (Geospatial), RMIT University, Melbourne, VIC 3001, Australia; haobo.li@rmit.edu.au
* Correspondence: wxm@aoe.ac.cn; Tel.: +86-10-82178896

**Abstract:** In this study, we investigate the performances of a commercial Global Navigation Satellite System (GNSS) Radio Occultation (RO) mission and a new-generation RO constellation, i.e., Spire and Constellation Observing System for Meteorology, Ionosphere, and Climate 2 (COSMIC-2), respectively. In the statistical comparison between Spire and COSMIC-2, the results indicate that although the average signal-to-noise ratio (SNR) of Spire is far weaker than that of COSMIC-2, the penetration of Spire is comparable to, and occasionally even better than, that of COSMIC-2. In our analysis, we find that the penetration depth is contingent upon various factors including SNR, GNSS, RO modes, topography, and latitude. With the reanalysis of the European Centre for Medium-Range Weather Forecasts and Radiosonde as the reference data, the identical error characteristics of Spire and COSMIC-2 reveal that overall, the accuracy of Spire's neutral-atmosphere data products was found to be comparable to that of COSMIC-2.

**Keywords:** GNSS-RO; Spire; COSMIC-2

## 1. Introduction

For nearly two decades, Global Navigation Satellite System (GNSS) Radio Occultation (RO) has served as a remote sensing technique providing vast amounts of data for numerical weather prediction [1,2], space weather analyses [3], and climate change research [4,5]. During GNSS-RO measurements, spaceborne receivers on low-Earth-orbiting (LEO) satellites collect GNSS signals affected by refraction from the Earth's atmosphere when GNSS satellites disappear or emerge past the Earth's horizon, yielding atmospheric profiles with the advantages of high accuracy, high vertical resolution, all-weather capability, and global coverage [6,7].

With the successful GPS/MET [8] experiment, the GNSS-RO technique has been identified as a promising technique for the retrieval of atmospheric profiles. The GPS/MET experiment, launched in April of 1995, played a pivotal role in demonstrating the feasibility and effectiveness of GNSS-RO for atmospheric research. Thus, atmospheric monitoring research of Earth involving GNSS-RO observations has been in demand, contributing to a series of RO missions, including the Ørsted [9], Sunsat [10], Challenging Minisatellite Payload [11], Satellite for Scientific Applications C/D [12], Gravity Recovery and Climate Experiment [13], Constellation Observing System for Meteorology, Ionosphere, and Climate 1/2 (COSMIC-1/2) [14,15], Meteorological Operational satellite program-A/B/C [16], and

Feng Yun-3C/D [17] missions. Notably, COSMIC-2 is a joint mission launched by Chinese Taiwan and the U.S. involving multiple satellites designed to collect RO data using signals from GNSS. The mission aims to enhance global weather prediction, ionospheric research, and climate monitoring. Although the number of RO profiles produced by operational GNSS-RO missions is currently far below the minimum profile number required for global observing systems, notably, a frequency of 16,000–20,000 globally distributed occultations per day can achieve the requirement of assimilation in numerical weather prediction [18]. Recently, due to the advantages of small-satellite technologies, including their low economic costs and short research and development periods, RO techniques have been rapidly developed; thus, some commercial GNSS-RO missions have been performed with small satellites to complement the shortage of scientific GNSS-RO data, such as Spire [19]. Spire is a commercial space-based company that operates a constellation of nanosatellites for various purposes, including RO measurements. In the Spire program, a constellation of nanosatellites, known as the LEO Multi-Use Receiver Satellite Bus, has been built to generate a tremendous amount of RO data.

Many evaluations of GNSS-RO data have been implemented to verify the high quality of GNSS-RO retrievals, including GPS/MET [8], Challenging Minisatellite Payload [20], Gravity Recovery and Climate Experiment [13], Meteorological Operational satellite program [21], COSMIC-1 [14], etc. However, comprehensive investigations into the parallels and distinctions between scientific occultation missions and commercial nanosatellite occultation missions, such as COSMIC-2 and Spire, have been limited. Ho et al. (2020) [15] preliminarily investigated the COSMIC-2 neutral atmospheric profile quality using radiosonde data and other RO profiles in terms of profile precision, stability, accuracy, and uncertainty. They found that COSMIC-2 data had a relatively consistent quality with that of COSMIC-1, and the higher signal-to-noise ratio (SNR) of the COSMIC-2 measurements allowed RO signals to penetrate deeper into the lower troposphere, slightly improving the retrieval accuracy. Chen et al. (2021) [22] made a statistical comparison of COSMIC-2 with data from radiosonde, RO data from other missions, global analyses from ECMWF and the National Centers for Environmental Prediction final, and other satellite products, and all the datasets had consistent vertical variations. The temperature profiles showed a mean difference of <0.5 °C and a standard deviation (STD) of 1.5 °C, and the water vapor pressure showed deviations within 2 hPa in the lower troposphere. Spire has operated a constellation of more than 110 LEO nanosatellites since 2019, and Spire's RO refractivity profiles have comparable quality with that of COSMIC-2 [23]. Johnston et al. (2021) conducted a comparison between specific humidity profiles derived from COSMIC-2 RO data and those from ERA5 and MERRA-2 reanalysis datasets. The findings reveal a strong concordance between COSMIC-2 specific humidity and ERA5 while highlighting more pronounced discrepancies with MERRA-2, especially within the boundary layer [24]. Forsythe (2020) et al. validated the ionospheric electron density through Spire's CubeSats RO measurements, and the RO ionospheric inversion results showed significant consistency with the digisonde measurements and Arecibo incoherent scatter radar data [25].

Although the studies above already estimated the RO profiles of Spire and COSMIC-2 and obtained some preliminary results, the properties of the Spire and COSMIC-2 RO retrievals, such as their global coverage and SNR influence, discrepancies between their retrieval qualities, have not been studied. The GNSS-RO constellation observation distribution exhibits global coverage [26], and the SNR is the critical factor limiting the deeper penetration of the GNSS-RO observations [27]. Additionally, the limited payloads of the small satellites result in low power consumption and low gain antennas, thereby reducing the RO retrieval quality. In this study, we aimed to systematically analyze Spire and COSMIC-2 RO profiles from UCAR with other datasets, including ECMWF Reanalysis and radiosonde datasets.

This paper is organized as follows. An introduction to the data and methodology is given in Section 2. The systematic comparison of the Spire and COSMIC-2 retrievals with

the ECMWF reanalysis and radiosonde data is discussed in Section 3. Finally, conclusions are provided in Section 4.

## 2. Data and Methodology

### 2.1. GNSS-RO Data

The Spire and COSMIC-2 RO data involved in this study, including the neutral atmospheric excess phase and "wet" profile products, are available to freely download from the COSMIC Data Analysis and Archive Center (CDAAC) (https://www.cosmic.ucar.edu/ accessed on 21 October 2023) [28]. The SNRs of RO events are recorded in the excess phase files and meteorological parameters, such as the refractivity, pressure, temperature, relative humidity, specific humidity, and water vapor pressure, are provided in the "wet" profile files [15]. In the current study, Spire and COSMIC-2 RO data from the day of year (DOY) 60 in 2022 to 059 in 2023 are used. It is important to emphasize that the Spire data comprises three navigation satellite systems: GPS, GLONASS, and GALILEO, whereas the COSMIC-2 data only includes GPS and GLONASS.

The equation representing refractivity ($N$), which is a function of pressure ($P$ in hPa or mbar), temperature ($T$ in K), and water vapor pressure ($e$ in hPa or mbar) in the neutral atmosphere, is given as follows by Smith et al. (1953) [29].

$$N = 77.6\frac{P}{T} + 3.73 \times 10^5 \frac{e}{T^2}. \tag{1}$$

Based on the assumption that water vapor could be negligible, RO "dry" profiles, including "dry pressure" and "dry temperature", are obtained by Equation (1). However, this assumption is unreasonable because more moisture exists below the upper troposphere. Hence, RO "wet" profiles (level-2 products, named "wetPrf" or "wetPf2") including moisture information are extracted using the one-dimensional variational (1DAR) method from the RO bending angle profiles [30]. The vertical resolution of the "wet" profiles is 0.05 km from the surface to below 20 km altitude and 0.1 km from above 20 km to 60 km altitude.

### 2.2. ERA5 Datasets

ERA5 is the fifth-generation global atmospheric reanalysis product [31], and hourly ERA5 data representing pressure levels during 2019, used in this study, are among the most advanced three-dimensional global analyses available for estimating the quality of Spire and COSMIC-2 RO profiles as benchmark values. As shown in Table 1, the required variables in the ERA5 dataset related to this study, including the specific humidity (kg/kg), temperature (K), and geopotential ($m^2/s^2$), are available at a horizontal resolution of $0.25° \times 0.25°$ on 37 pressure levels from 1000 hPa to 1 hPa. It can be downloaded publicly from the provided URL https://cds.climate.copernicus.eu/cdsapp#!/dataset/reanalysis-era5-pressure-levels?tab=form (accessed on 21 October 2023).

**Table 1.** ERA5 hourly data on pressure levels used in this study.

| Projection | Regular Latitude-Longitude Grid |
|:---:|:---:|
| Horizontal coverage | Global |
| Horizontal resolution | $0.25° \times 0.25°$ |
| Vertical coverage | 1000 hPa to 1 hPa |
| Vertical resolution | 37 pressure levels |
| Temporal resolution | Hourly |
| Required variables | Specific humidity, temperature, and geopotential |

### 2.3. Radiosonde Data

The Integrated Global Radiosonde Archive version 2 (IGRA2) provided by the National Centers for Environment Information is a radiosonde dataset [32] containing variables such as pressure, geopotential height, temperature, and relative humidity from high-quality sounding performed at more than 2800 globally distributed stations; these data

are accessible at the website https://www.ncei.noaa.gov/pub/data/igra/ (accessed on 21 October 2023). Here, IGRA2 data are used as the other benchmark values to eliminate the effect of the assimilation of COSMIC-2 and Spire data in the ECMWF Integrated Forecasting System since March and May of 2020, respectively [33]. The IGRA2 observation data are very limited above the 30 km altitude due to the flight limits of radiosondes. Hence, the limit height for comparison between the GNSS-RO and radiosonde data is set to 30 km in this work.

*2.4. Methodology*

In the comparison of radiosonde and GNSS-RO data, data pairs are collocated within the spatiotemporal windows of 1 h and 100 km. Furthermore, the vertical resolutions of both the ERA5 and radiosonde data are not comparable to that of the GNSS-RO data. Therefore, the ERA5 and radiosonde data are interpolated into the vertical resolutions of the GNSS-RO data.

In this study, the mean difference and STD used in the statistical calculations are defined using the following equations to evaluate the GNSS-RO product properties:

$$\overline{\Delta x_a} = \frac{1}{n}\sum_{i=1}^{n}\left(x_i^{ro} - x_i^t\right), \tag{2}$$

$$STD_{\Delta x_a} = \sqrt{\frac{1}{n}\sum_{i=1}^{n}\left(\left(x_i^{ro} - x_i^t\right) - \overline{\Delta x_a}\right)^2}, \tag{3}$$

$$\overline{\Delta x_r} = \frac{\frac{1}{n}\sum_{i=1}^{n}\left(x_i^{ro} - x_i^t\right)}{x_i^t}, \tag{4}$$

$$STD_{\Delta x_r} = \sqrt{\frac{1}{n}\sum_{i=1}^{n}\left(\frac{\left(x_i^{ro} - x_i^t\right)}{x_i^t} - \overline{\Delta x_r}\right)^2}, \tag{5}$$

where $x_i^{ro}$ and $x_i^t$ represent the GNSS-RO and benchmark temperature, relative humidity, pressure, and refractivity, respectively, the subscript $i$ stands for the $i$th GNSS-RO-benchmark collocation, and $n$ is the number of collocations.

Emphasis should be placed on the fact that data quality control is conducted as part of the data quality assessment process. The reference values were derived from ERA5 or radiosonde data. RO refractivity profiles with relative errors surpassing 5%, as well as wet pressure profiles exceeding 900% or dropping below −90%, were eliminated.

**3. Comparison Results**

*3.1. Initial Analysis*

The GNSS-RO constellation pattern impacts the distribution of RO event observations over the globe. Without the specific configuration for Spire, consisting of a diverse set of orbits compounding Sun-Synchronous Orbits (SSO), 83–85° Orbits, Equatorial Orbit, 51.6° Orbits, and 37° Orbits and the continuous changing of satellites owing to their short operational lifetime of 2+ years [34]. As for COSMIC-2, six satellites orbit around the Earth at an altitude of 550 km with a 24° inclination [35]. Furthermore, given that GNSS constellations encompass diverse signal frequencies, constellation configurations, and modulation-demodulation techniques, potentially influencing RO events. Hence, separate investigations will be conducted for GPS, GLONASS, and GALILEO RO events.

The spatial distribution of the Spire RO events obtained from satellites with a different orbit type has an obvious difference. As shown in Figure 1, the Spire RO events observed on DOY 60 in 2022 are scattered globally. The red dots signify GPS RO events, the green dots indicate GLONASS RO events, whereas the blue dots represent GALILEO RO events. For COSMIC-2, Chen et al. (2021) [22] mention that RO events only cover the low-latitude area (±45°). Notably, few RO events recorded by COSMIC-2 occurred slightly beyond the

edges of the ±45° region; these events are regarded as occurring on the scale within the latitude area of ±45° in this study. The coverage areas of the RO events by Spire are wider than those by COSMIC-2; thus, Spire can provide global RO data due to the constellation characteristics of LEMUR-2 consisting of a series of orbits.

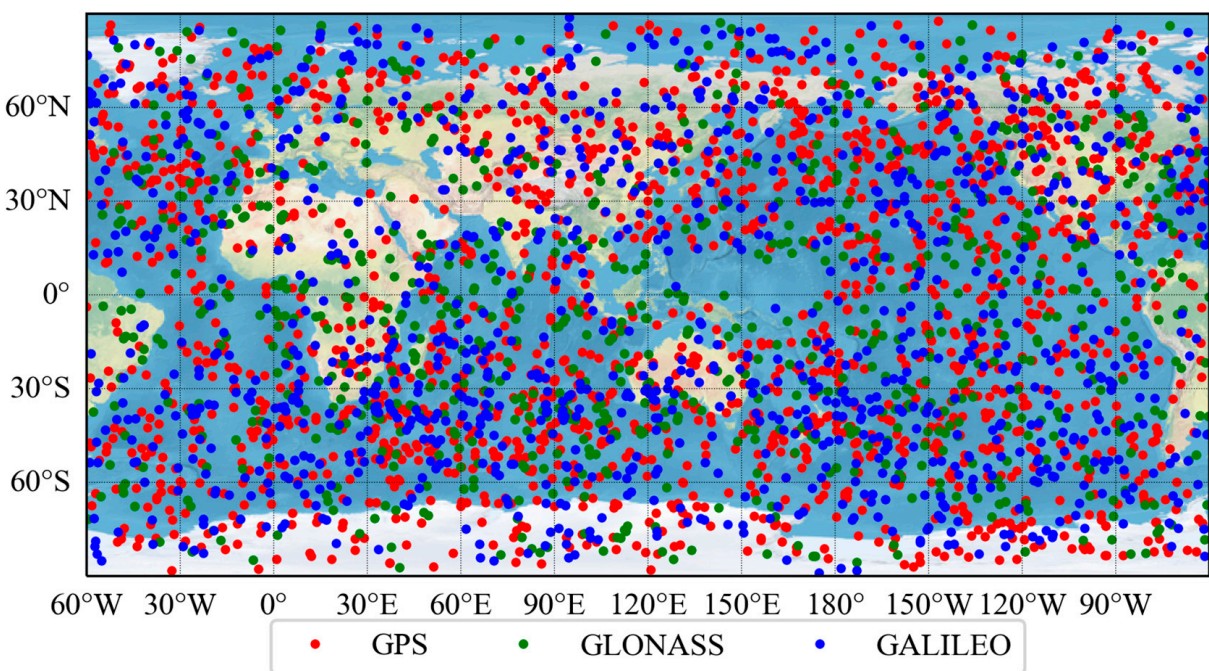

**Figure 1.** Spire RO event distribution on DOY 060 in 2022 (red: GPS, green: GLONASS, and blue: GALILEO).

Furthermore, the RO events can be classified into two modes, setting occultation or rising occultation, based on the relative movement trend on the occultation observation. Considering the different navigation satellite systems, here, the RO events were divided into six groups: GPS/Set, GPS/Rise, GLONASS/Set, GLONASS/Rise, GALILEO/Set, and GALILEO/Rise.

Table 2 shows the percentage for six groups of profiles (wetPf2) from the Spire and COSMIC-2 satellite data obtained from CDAAC from DOY 060 in 2022 to 059 in 2023 under quality control. Generally, there were more setting occultation events than rising occultation events (except GLONASS-derived RO events for Spire). The number of atmospheric profiles for Spire amounts to 1,663,197, surpassing COSMIC-2 with the number of 1,440,424, which is a great supplement to the high-latitude regions that COSMIC-2 data cannot cover. However, the Spire satellite constellation comprises dozens of satellites, whereas COSMIC-2 consists of only six. Daily data produced by individual Spire satellites from UCAR contain no more than 300 RO soundings, which is much smaller than those recorded by individual COSMIC-2 satellites, each providing over 700 RO soundings.

**Table 2.** The percentage for six groups of profiles (wetPf2) from the Spire and COSMIC-2 data obtained from CDAAC from DOY 060 in 2022 to 059 in 2023 under quality control.

| GNSS-RO Mission | Number of Profiles | Mode | GPS | GLONASS | GALILEO |
|---|---|---|---|---|---|
| Spire | 1,663,197 | Set | 26.46% | 14.80% | 12.14% |
|  |  | Rise | 19.97% | 15.29% | 11.34% |
| COSMIC-2 | 1,440,424 | Set | 34.08% | 18.71% | None |
|  |  | Rise | 29.43% | 17.78% | None |

With the RO events observed from DOY 060 in 2022 to 059 in 2023 by Spire and COSMIC-2, we carried out an investigation on the RO events obtained with different navigation satellite systems, i.e., GPS, GLONASS, and GALILEO. As shown in Table 2, the percentages of the GPS-derived RO events are much larger than GLONASS- and GALILEO-derived RO events for both Spire and COSMIC-2. This is reasonable considering the number of operational GPS satellites is larger than GLONASS and GALILEO satellites. For Spire, GLONASS-derived RO events are slightly higher than those derived from GALILEO. Although there is no significant difference in the number of satellites in orbit for GLONASS and GALILEO, it is important to note that some Spire satellites do not receive GALILEO signals, including S128, S115, S117, and others.

*3.2. SNR*

The SNR is the key factor impacting deeper occultation, especially for surface and tropical RO soundings. Currently, Spire and COSMIC-2 have developed new-generation GNSS-RO payloads, i.e., STRATOS and the Tri-GNSS Radio-occultation System (TGRS), respectively, to improve their retrieval quality in the low troposphere. It should be noted that, compared to other payloads with high-gain antennas (e.g., the TGRS and CION [27]), STRATOS is equipped with relatively low-gain antennas to track GNSS signals, thus directly leading to relatively low SNR values in their measurements.

Figure 2 features a dual *x*-axis. The top *x*-axis represents the altitude in km, being the straight-line height between the GNSS and LEO satellites, while the bottom *x*-axis represents the time sequence in seconds. The *y*-axis corresponds to the SNR for two GPS frequency bands. In this context, there are two GPS frequency bands, L1 at 1575.42 MHz and L2 at 1227.60 MHz. In Figure 2, the SNR time series of two examples of a typical rising occultation, observed in an adjacent area in the tropics nearly simultaneously by STRATOS and the TRGS, are depicted. The 1-s average SNR of the two rising occultation events shown with the black and green lines increases with increasing altitude. Furthermore, obvious fluctuations or oscillations in the SNR curve can be observed at altitudes between $-100$ km and $-50$ km, resulting from signals being temporarily captured and then abruptly disappearing due to atmospheric ducting and super refraction [12]. It appears that the SNR time series of the Spire is lower than that of the COSMIC-2. For example, at altitudes above 0 km, the L1 SNR of the Spire is ~320 volts/volt, approximately one quarter of the L1 SNR of the COSMIC-2, which is ~1400 volts/volt. Similarly, the L2 SNR of the Spire is ~200 volts/volt, nearly half of the L2 SNR of the COSMIC-2, which is ~500 volts/volt. Moreover, according to the study by Sokolovskiy et al. (2014) [36], the L1 and L2 SNRs of the COSMIC-1 during several rising occultation events were ~600–800 volts/volt and ~200–600 volts/volt, respectively, at altitudes above 0 km. Therefore, the results that STRATOS, which has relatively low-gain antennas, has a slightly weaker ability to capture and track signals compared to IGOR, the payload of the COSMIC-1. Also, TRGS has an even stronger ability to capture and track signals compared to both STRATOS and IGOR.

To generally compare the capability of STRATOS and the TRGS in capturing and tracking signals, we also investigated the SNR in the altitude range of 60–80 km. The 60–80 km altitude range is optimal for evaluating signal strength, unaffected by atmospheric interference. It's sufficiently high to render attenuation from typical atmospheric refraction negligible, yet it doesn't extend to the E-layer where disturbances are more pronounced [21,27]. As a result, the average L1-signal SNR within the 60–80 km altitude range (hereafter referred to as the SNR average) is related to the signal strength of the RO event. Figure 3 shows the average SNR histograms of the normalized probabilities for the Spire, while Figure 4 displays the average SNR histograms of the normalized probabilities for the COSMIC-2. For both GPS and GLONASS, the SNR averages of all the Spire data (blue) range from ~200–1600 volts/volt with only one peak, which is much weaker than the COSMIC-2 averages (brown), which vary from ~200–2250 with two peaks (see Figures 3a and 4a). To investigate the influence of the navigation satellite systems and occultation mode (setting or rising occultation) received by LEO on the SNR, all the

data were divided into six groups: GPS/Set, GPS/Rise, GLONASS/Set, GLONASS/Rise, GALILEO/Set, and GALILEO/Rise.

In Figure 3b,c and Figure 4b,c, the tops and bottoms of the x-axes represent the SNR averages of the setting and rising occultation-normalized probability histograms, respectively. Both the GPS/Set and GPS/Rise SNR averages of the Spire range from ~200–600 volts/volt, while those of COSMIC-2 range from ~300–2000 volts/volt. In Figure 3c, the SNR averages for both the GLONASS/Set and GLONASS/Rise of the Spire range from ~300 to 1500 volts/volt, while those of COSMIC-2 range from ~250–2500 volts/volt in Figure 4c. In Figure 3d, the SNR averages for both the GALILEO/Set and GALILEO/Rise of the Spire range from ~300 to 1500 volts/volt. The SNR averages for both the GALILEO/Set and GALILEO/Rise of the Spire span from ~300 to 750 volts/volt. It is possible that the broader range of the SNR averages for GPS- or GALILEO-derived RO data is due to its utilization of CDMA wireless data transmission, while GLONASS utilizes FDMA. It's worth noting that the occultation mode does not affect the scale of the SNR averages, while the navigation satellite system (either GPS, GLONASS, or GALILEO) has an outstanding effect on the range of the SNR averages. The SNR averages of the GPS-derived RO exhibit a sharper peak compared to that of the GLONASS-derived RO, as shown in the comparison between Figure 3b,c. The GPS- and GLONASS-derived RO data of the Spire produce two distinct peaks that are widely separated from each other, as shown in Figure 3a. In contrast, Figure 4a displays only one peak due to the proximity of the COSMIC-2 peaks, while the SNR averages for GALILEO-derived RO data do not exhibit a clear peak in Figure 4c.

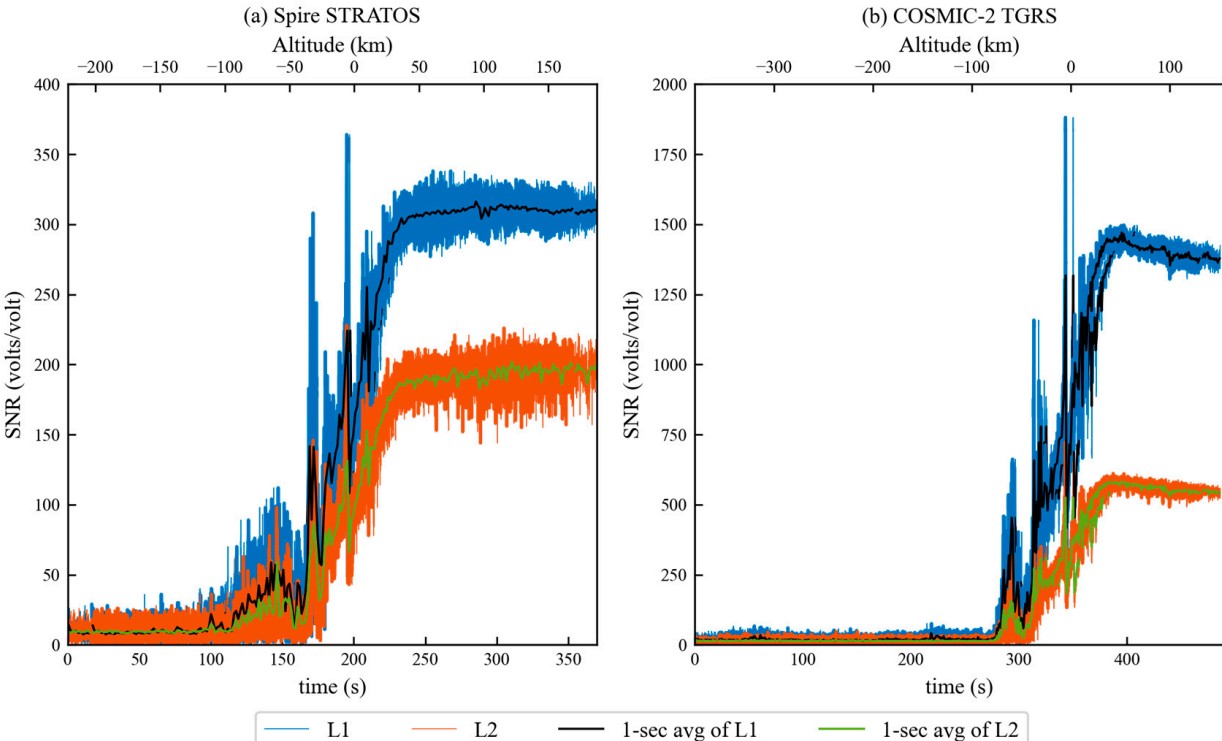

**Figure 2.** SNRs of two examples of a typical rising occultation in a tropical region with open-loop tracking (blue: L1, orange: L2, black: 1-s average of L1, and green: 1-s average of L2). (**a**) STRATOS, RO starting time 1847 UTC, 1 March 2022, located at 29.79°N, 141.45°W; (**b**) the TGRS, RO starting time 1845 UTC, 1 March 2022, located at 30.52°N, 142.44°W. (The top *x*-axis represents the altitude in km, being the straight-line height between the GNSS and LEO satellites, while the bottom *x*-axis represents the time sequence in second. The *y*-axis corresponds to the SNR for two GPS frequency bands).

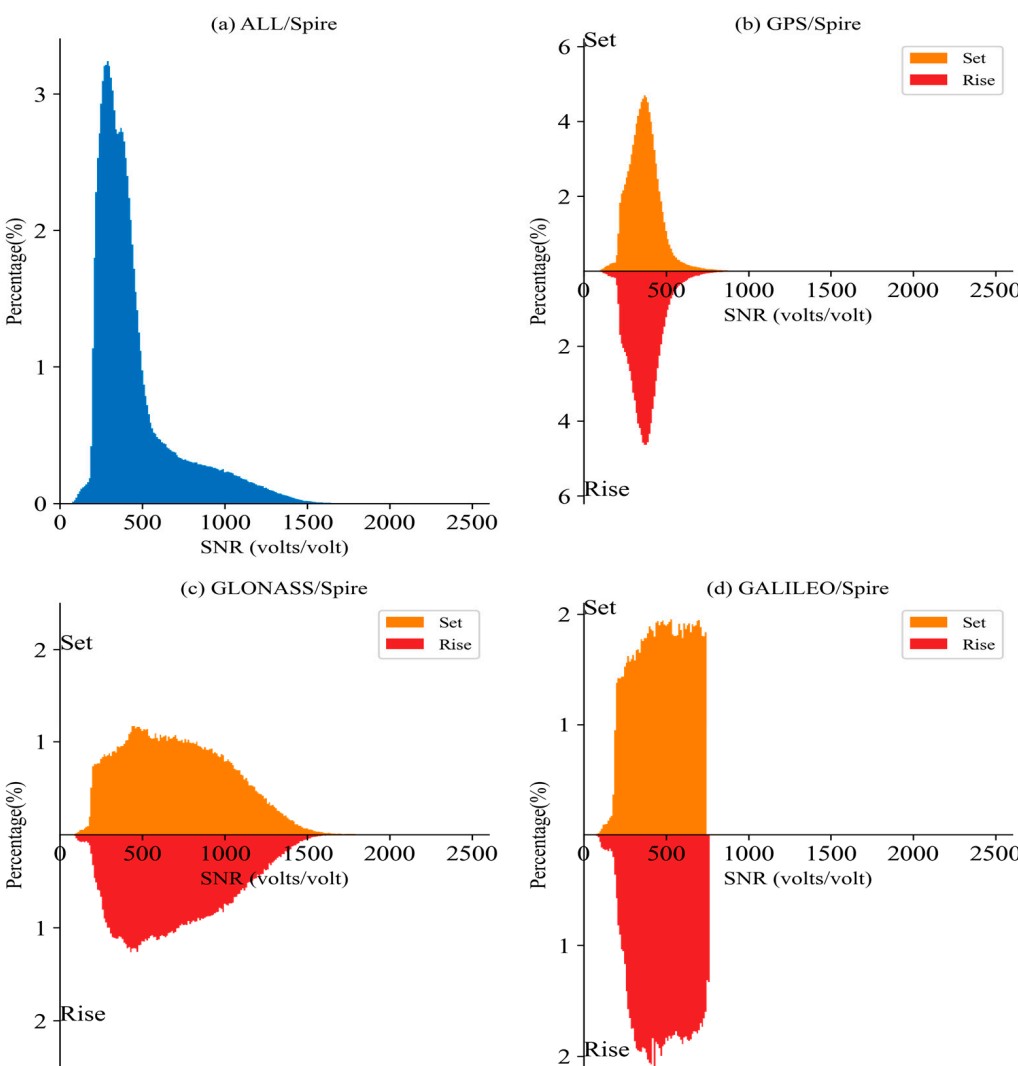

**Figure 3.** Histograms of the normalized probability of the average L1 SNR (volts/volt) values for the Spire between 60 and 80 km.

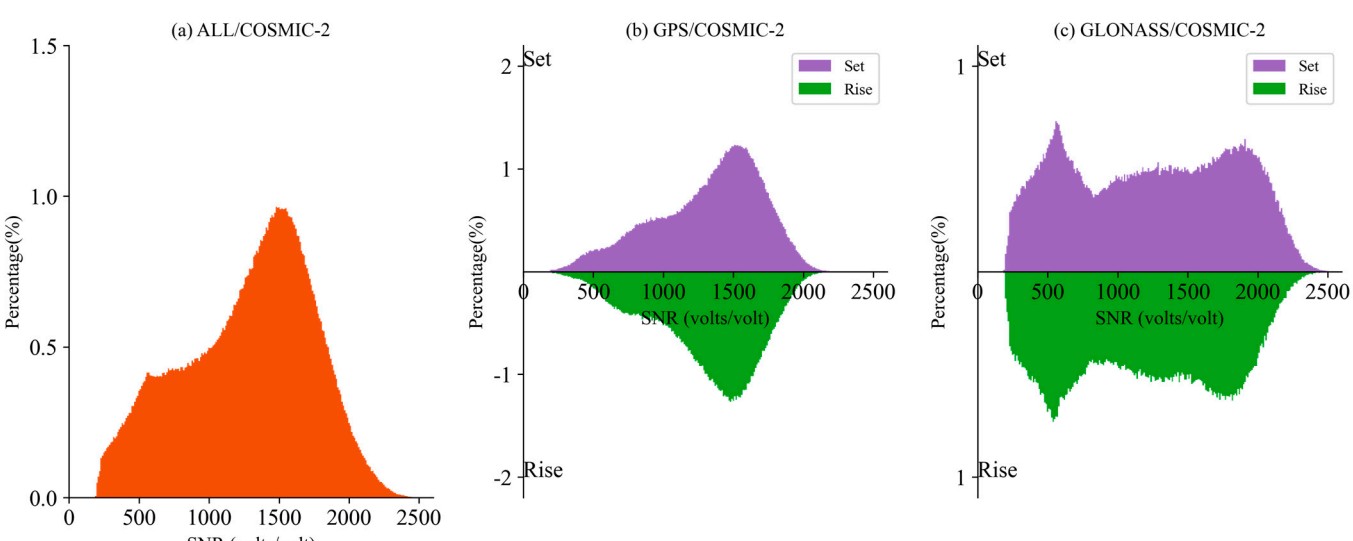

**Figure 4.** Histograms of the normalized probability of the average L1 SNR (volts/volt) values for the COSMIC-2 between 60 and 80 km.

Ho et al. (2020) mentions that enhancing SNR can improve penetration and data quality [15], and Jing et al. (2023) points out the correlation between the penetration of COSMIC-2 and latitude [37]. Therefore, based on these insights, to analyze the correlations between the SNR and penetration and between the SNR and data quality, it was necessary to compare the mean SNRs at different latitudes. Thus, a statistical comparison was performed on all the data and for the four data groups at latitudinal intervals of 15°. As shown in Figure 5, the influence of the RO mode on the mean SNR with latitudinal variations was not significant. Figure 5a shows that the mean SNR values of the GPS- and GALILEO-derived RO data of the Spire do not show a clear dependence on latitude, while the mean SNR values of the GLONASS-derived RO data of the Spire show some fluctuation with latitude. For COSMIC-2, Figure 5b shows that generally, the mean SNR values are much higher in the low latitudes than in the high latitudes.

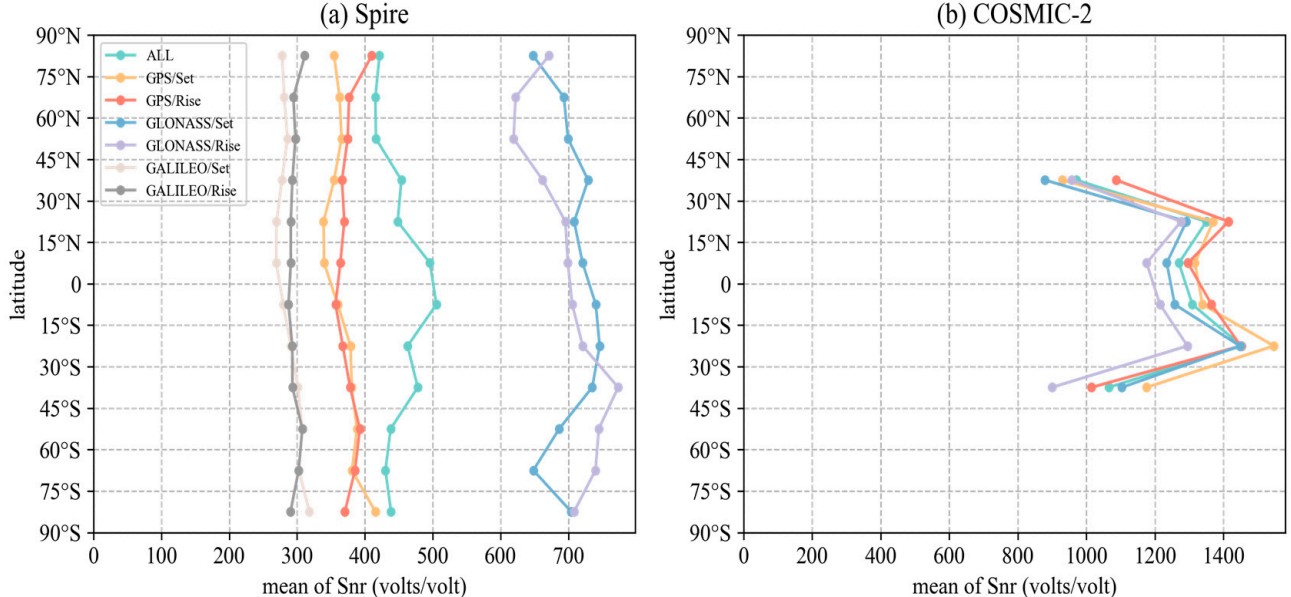

**Figure 5.** Mean SNR values of the Spire and COSMIC-2 at different latitudes: (**a**) Spire and (**b**) COSMIC-2. Light green: all data, orange: GPS/Set, Red: GPS/Rise, blue: GLONASS/Set, and purple: GLONASS/Rise, khaki: GALILEO/Set, and gray: GALILEO/Rise.

In summary, the mean SNR values of the Spire were significantly smaller than that of the COSMIC-2. In addition, the mean SNR values of the GPS-derived RO showed a sharper peak for both the Spire and COSMIC-2 than mean SNR values of the GLONASS-derived RO, with no effect of the occultation mode observed. Notably, the SNR averages of the GALILEO-derived RO data show no obvious peak. We then examined the penetrations of different missions in relation to the varying SNR strengths in Section 3.3.

Table 3 presents the mean SNR values for the Spire, COSMIC-2, and COSMIC-1. The mean SNR values of the Spire for the GPS-, GLONASS, and GALILEO-derived RO were 371, 708, and 480 volts/volts, respectively, and the total mean SNR for Spire was 503 volts/volts. As for the COSMIC-2, the mean SNR values for the GPS- and GLONASS-derived RO were 1315 and 1210 volts/volt, respectively, and the total mean SNR for COSMIC-2 was 1276 volts/volt. The mean SNR value for the COSMIC-1, obtained from [38], was 704 volts/volt. These results indicate that the ability of the Spire to track only GPS signals is significantly weaker than that of the COSMIC-1, and with the joint consideration of GLONASS and GALILEO, Spire can achieve a slightly weaker capability than COSMIC-1. The COSMIC-2 has a superior ability to track signals compared to both the COSMIC-1 and Spire.

**Table 3.** Mean L1 SNR values of Spire, COSMIC-2 and COSMIC-1 (unit: volts/volt).

| GNSS-RO Mission | GPS | GLONASS | GALILEO | Total |
|---|---|---|---|---|
| Spire | 371 | 708 | 480 | 503 |
| COSMIC-2 | 1315 | 1210 | None | 1276 |
| COSMIC-1 | 704 | None | None | 704 [38] |

*3.3. Penetration*

The lower atmosphere is important for numerical weather prediction and atmospheric science research. Because of thick water vapor near the surface, RO observations are limited in their ability to provide atmospheric information in the lower troposphere. As a result, the penetration GNSS-RO detected is also an important indicator of the quality of RO sound. In this section, statistical analysis was conducted on the penetration of COSMIC-2 and Spire, and the effect of SNR on the penetration was analyzed.

As shown in Table 4, 76.60% of the Spire data can achieve penetration depths below 1 km in the lower troposphere. The COSMIC-2 was able to detect the surface atmosphere at a 1 km height in about 78.12% of RO events, which is better than the Spire. According to Ho et al. (2020), increasing the SNR can improve the RO penetration depth, which improves the ability of COSMIC-2 in penetrating deep into the lowest 100 m of the troposphere. Therefore, the conclusion that the COSMIC-2 can penetrate deeper than the Spire is consistent with previous experiments [15] that attribute this difference to the higher SNR of the COSMIC-2. Moreover, in comparison to the capability to detect the surface atmosphere at a 1 km height, it is noteworthy that setting occultation events can achieve greater penetration depths than rising occultation events for both Spire and COSMIC-2, independent of the navigation satellite system's influence.

**Table 4.** Penetration depth percentages of Spire and COSMIC-2 from DOY 060 in 2022 to 059 in 2023. (altitude of penetration $H_p$).

| GNSS-RO Mission | Group | $H_p \leq 1$ km | 1 km < $H_p \leq 5$ km | 5 km < $H_p \leq 10$ km | $H_p > 10$ km |
|---|---|---|---|---|---|
| Spire | GPS/Set | 78.62% | 19.75% | 1.55% | 0.08% |
| | GPS/Rise | 77.38% | 21.08% | 1.46% | 0.09% |
| | GLONASS/Set | 78.29% | 20.16% | 1.49% | 0.06% |
| | GLONASS/Rise | 74.31% | 23.93% | 1.64% | 0.12% |
| | GALILEO/Set | 79.05% | 19.29% | 1.56% | 0.10% |
| | GALILEO/Rise | 73.29% | 24.57% | 2.02% | 0.12% |
| | Total | 76.60% | 22.20% | 1.07% | 0.13% |
| COSMIC-2 | GPS/Set | 79.20% | 19.87% | 0.90% | 0.03% |
| | GPS/Rise | 75.34% | 23.61% | 1.01% | 0.04% |
| | GLONASS/Set | 78.21% | 20.57% | 1.02% | 0.21% |
| | GLONASS/Rise | 75.09% | 23.62% | 1.21% | 0.08% |
| | Total | 78.12% | 20.79% | 1.01% | 0.08% |

However, it is important to note that the statistical method used above is not perfect due to the influence of topography (e.g., mountains) on the penetration depth. Taking terrain into account (using data from ETOPO2 v2), we plotted the Spire RO events in a global topographic map where areas at $H_t \leq 1$ km, 1 km < $H_t \leq 5$ km, and $H_t > 5$ km (terrain high $H_t$) are denoted with white, brown, and deep brown colors, as shown in Figure 6. Figure 6 shows that RO events at $H_p \leq 1$ km (altitude of penetration $H_p$) are scattered in the region at $H_t \leq 1$ km (Figure 6a), those at 1 km < $H_p \leq 5$ km (Figure 6b) are found mainly in the region at 1 km < $H_t \leq 5$ km (e.g., the South Pole and mountainous areas), those at 5 km < $H_p \leq 10$ km (Figure 6c) are concentrated mainly in the region at $H_t > 5$ km (e.g., the

Himalayan Mountains and Andes Mountains), and those at $H_p > 10$ km (Figure 6d) are few and dotted around the world.

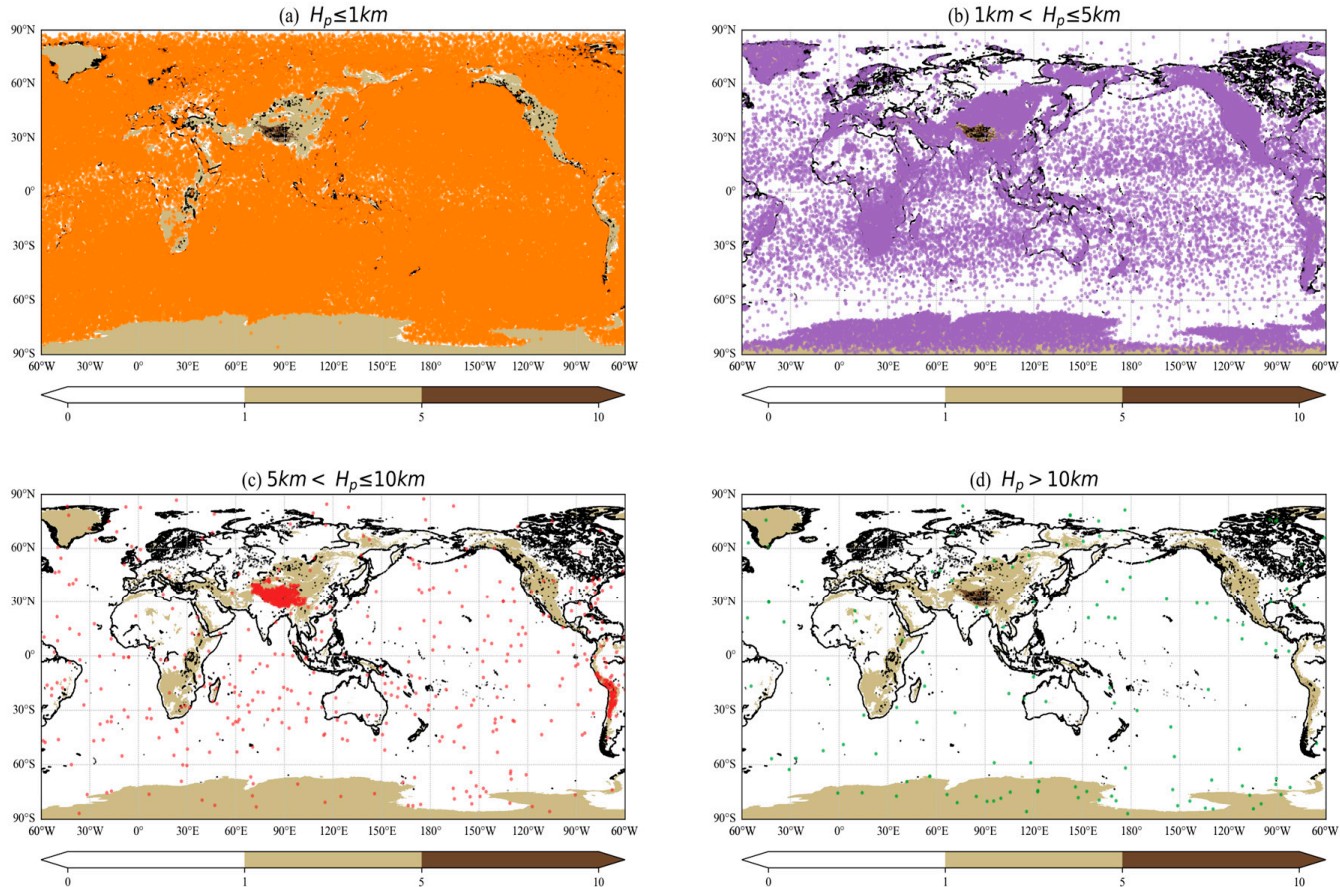

**Figure 6.** Penetration map of global-topography RO events recorded by Spire from DOY 060 in 2022 to 091 in 2022: (**a**) $H_p \leq 1$ km (altitude of penetration $H_p$), (**b**) 1 km $< H_p \leq 5$ km, (**c**) 5 km $< H_p \leq 10$ km, and (**d**) $H_p > 10$ km; white: $H_t \leq 1$ km, brown: 1 km $< H_t \leq 5$ km, and deep brown: $H_t > 5$ km (terrain high $H_t$).

Although the RO events recorded by the COSMIC-2 are limited to the lower-latitude area ($\pm 45°$) (as mentioned in Section 3.1), similar phenomena between the COSMIC-2 and Spire records are shown in Figure 7. Due to the great number of RO events recorded by COSMIC-2, the influences of topography are clearly visible, particularly for RO events at 5 km $< H_p \leq 10$ km, which are concentrated in the Himalayan Mountains and Andes Mountains, as indicated by the red points in Figure 7c.

Table 4 summarizes the percentages of RO data recorded below 1 km for the Spire and COSMIC-2, using the division scheme described in Section 3.1. The table shows that, in general, the percentages of the setting occultation events (except for GPS-derived RO events for the Spire, highlighted in gray) are slightly higher than those of the rising occultation events. Additionally, the ability to penetrate the lower troposphere during rising occultation events is slightly weaker than during setting occultation events. When compared to the COSMIC-2 data, the percentages of the corresponding items in the Spire data are higher, indicating that the Spire has a superior ability to penetrate the lower troposphere globally compared to the COSMIC-2 in lower-latitude regions. By considering the information in Table 4, the penetration depth also depends on the navigation satellite system and occultation mode.

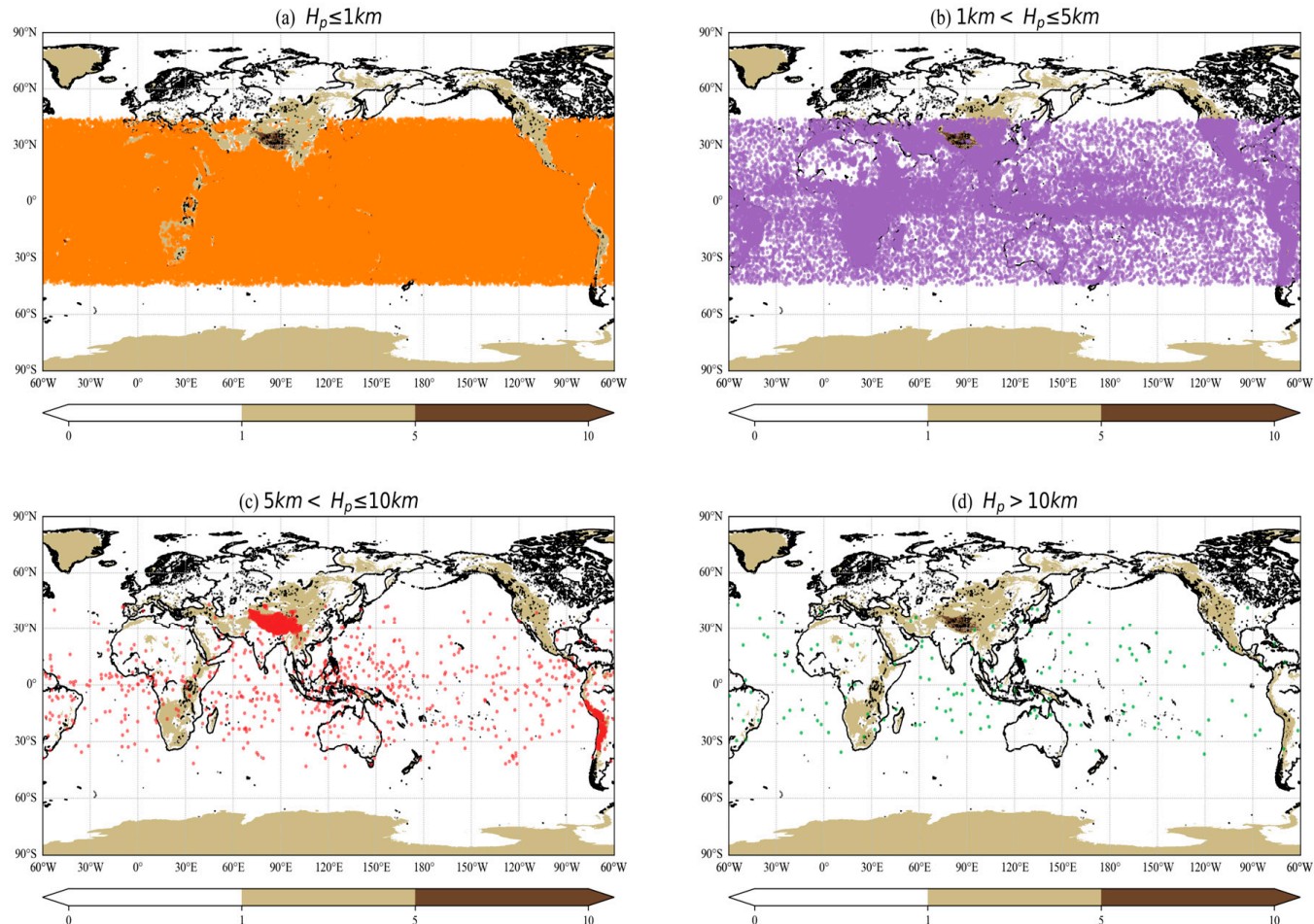

**Figure 7.** Penetration map of global-topography RO events recorded by COSMIC-2 from DOY 060 in 2022 to 091 in 2022: (**a**) $H_p \leq 1$ km (altitude of penetration $H_p$), (**b**) $1\,km < H_p \leq 5$ km, (**c**) $5$ km $< H_p \leq 10$ km, and (**d**) $H_p > 10$ km; white: $H_t \leq 1$ km brown: $1$ km $< H_t \leq 5$ km, and deep brown: $H_t > 5$ km (terrain high $H_t$).

To quantify the ability of the studied systems to detect the lower troposphere, the influences of topography were minimized by subtracting the terrain height from the penetration depth. Figure 8a shows the cumulative percentages of data below 1 km for the Spire and COSMIC-2 at different latitudes, indicating that the global Spire data are susceptible to topographic effects, particularly at the South Pole. With topographic effects accounted for, the COSMIC-2 surpasses the Spire within the latitude range of $\pm45°$. In general, the ability of the Spire global data, with topographic effects fixed, to penetrate the lower troposphere becomes stronger with increasing latitude. Combining Figures 5–7, the penetration depth is also determined by topography and latitude. Figure 8b exhibits the cumulative percentages of RO events at various penetration depths. The Spire RO events recorded with fixed topographic data show better performance in detecting the near-surface atmosphere, and similar results are seen for the COSMIC-2 data, which perform slightly worse than the fixed Spire data (see Figure 8).

After fixation, the Spire and COSMIC-2 data below an altitude of 1 km make up 88.7% and 85.3% of all the data, respectively. Due to topographic changes and water vapor variations with increasing latitude, the penetration depth is affected, thus leading to the retrieval statistics. Through comparing the fixed Spire data within the lower-latitude range of $\pm45°$ to the fixed COSMIC-2 data, it is found that the fixed Spire ($\pm45°$) data below the 1 km altitude accounted for 84.2% of all the data. In ascending sequence, the ability of

the systems to perform deeper soundings could be ranked as follows: Spire fixed (±45°), COSMIC-2 fixed, and Spire fixed.

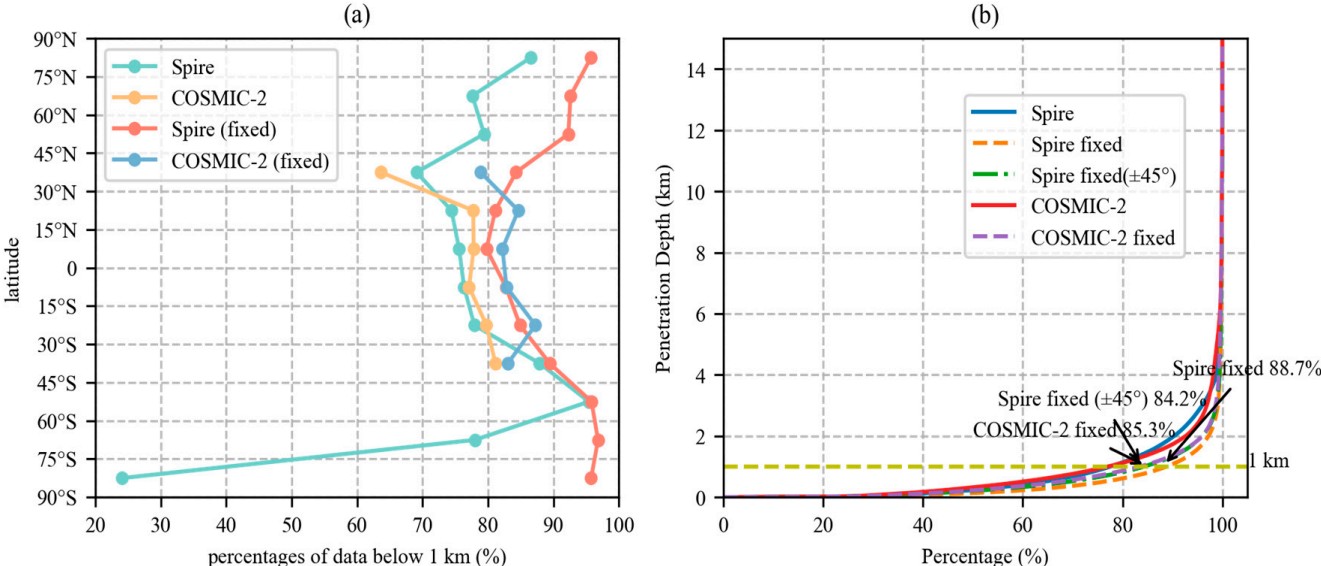

**Figure 8.** Cumulative percentages (**a**) of data below 1 km for Spire and COSMIC-2 at different latitudes (light green: Spire, orange: COSMIC-2, red: Spire with the topographic data fixed, and blue: COSMIC-2 fixed) and (**b**) cumulative percentages at each penetration for Spire and COSMIC-2 (blue solid line: Spire, orange dashed line: Spire fixed, green with dashed-dotted line: Spire fixed within the latitude area of ±45°, red solid line: COSMIC-2, and purple: COSMIC-2 fixed).

Therefore, COSMIC-2 has a better performance in sounding the deeper troposphere than Spire. Owing to topographic effects, especially the effects of mountains, the penetration depth is obviously affected. After removing the effects of terrain, the penetration of Spire is improved significantly, but under the same circumstances (e.g., at latitudes within ±45°), COSMIC-2 still has a better performance than Spire w.r.t deep penetration.

### 3.4. RO Retrieval Quality Assessment

The GNSS-RO product "wetPf2" includes atmospheric refractivity, temperature, specific humidity, and pressure. The other parameters are retrieved from the refractivity using the 1DAR method, so we preliminarily analyzed the refractivity characteristics. The COSMIC-2 RO data evaluated herein [15] have sufficient accuracy to assess the Spire data using stringent collocation criteria of a 1 h temporal window and a spatial distance of 100 km. Over 10 km, the differences were uniformly distributed on both sides of the *y*-axis at the zero point, as shown in Figure 9. Below 10 km, the differences between Spire and COSMIC-2 are largest, with more positive differences compared to the other heights, although these findings may have resulted from the increased water vapor or few considered collocation pairs.

Figure 10a shows that the root mean square (RMS) for the absolute difference of the refractivity below 30 km decreased with increasing latitude in comparison with the ERA5 dataset. In addition, the refractivity profiles of COSMIC-2 are chiefly in the tropics and diminish sharply as the latitude increases, while those of Spire are primarily situated in middle latitudes and few near the poles. Owing to the lower moisture at high latitudes, the Spire data was limited to the latitude range of ±45° to facilitate a comparison with the COSMIC-2 data.

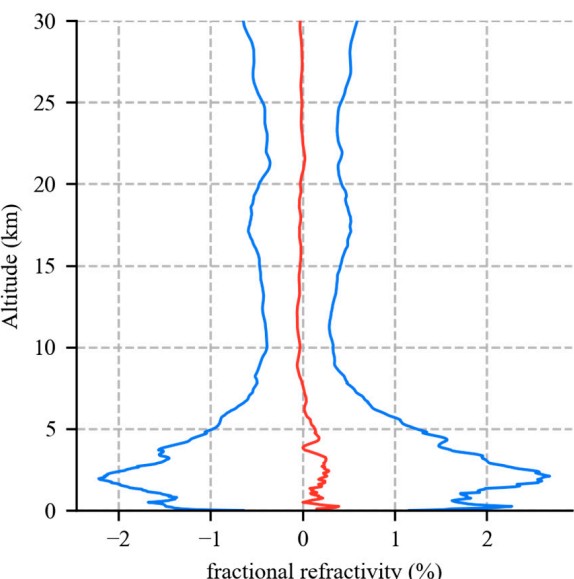

**Figure 9.** The fractional refractivity comparison between the Spire and COSMIC-2 wetPf2 data below 30 km (Red: mean fractional difference, and blue: STD).

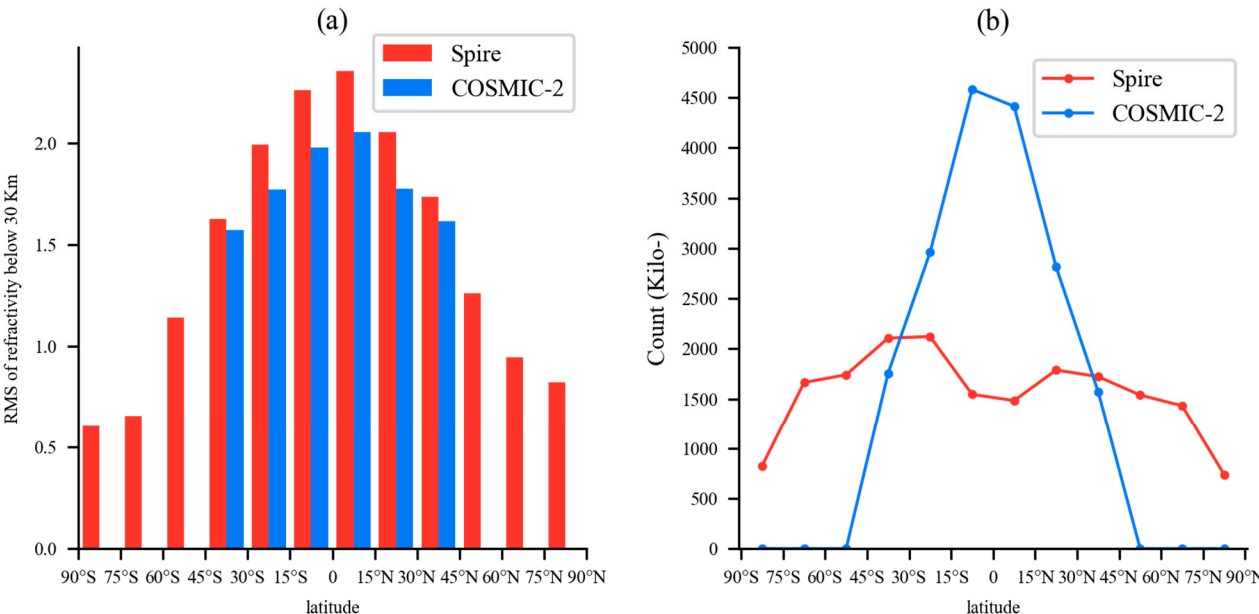

**Figure 10.** RMS of refractivity below 30 km and profile counts at different latitudes (red: Spire, blue: COSMIC-2). (**a**) RMS of refractivity below 30 km and (**b**) counts (Kilo-).

The evaluated GNSS RO data covers the entire year, enabling the analysis of the neutral atmospheric refractivity error seasonal characteristics below 30 km as shown in Figure 11. The solid lines represent the mean fractional differences in refractivity, while the dotted lines indicate the STD of fractional differences in refractivity. The black, red, green, and blue lines correspond to spring, summer, autumn, and winter, respectively. Figure 11a–c represent refractivity error seasonal characteristics for Spire and COSMIC-2 for Spire, Spire (±45°), and COSMIC-2, respectively. Based on the mean and STD of fractional differences in refractivity observed across different seasons, there does not appear to be any significant seasonal variation in refractivity errors. This may be due to the fact that GNSS uses L-Band navigation signals, which have the ability to penetrate clouds and rain, resulting in minimal weather-related interference.

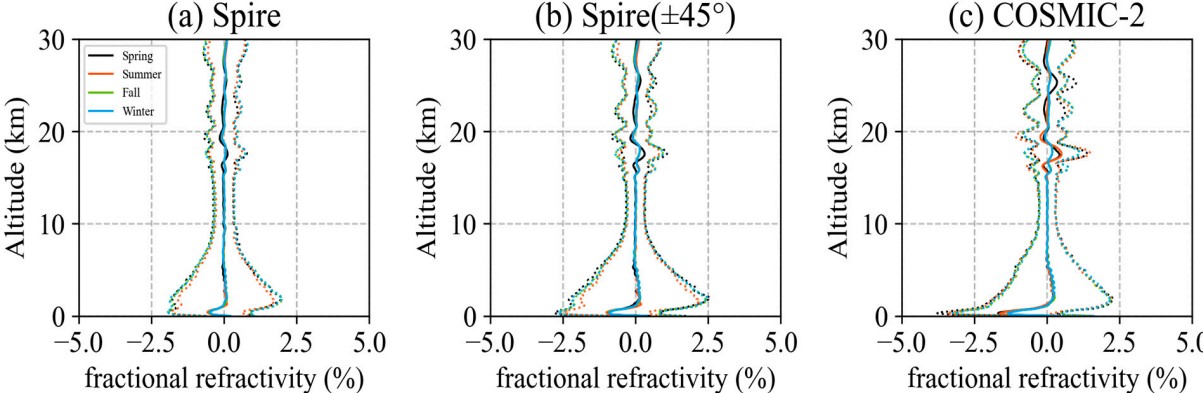

**Figure 11.** The neutral atmospheric refractivity error seasonal characteristics for Spire and COSMIC-2 from DOY 060 in 2022 to 059 in 2023 below 30 km. (black: Spring, red: Summer, green: Fall, and blue: Winter).

Due to the coarse coordinate scale in Figure 11 displayed, in Figure 12, the mean and STD of fractional differences in refractivity are presented separately to analyze the refractivity error characteristics. Also, it conducts an analysis of how different satellite navigation systems and RO event modes affect these refractivity error characteristics.

The neutral-atmosphere refractivity data acquired from Spire and COSMIC-2 were compared with the ERA5 dataset under the division scheme described in Section 3.1. The results obtained for all the data and for the six data groups are plotted in Figure 12. The statistics of those four groups (GPS/Set, GPS/Rise, GLONASS/Set, and GLONASS/Rise) have identical results, while GALILEO/Set and GALILEO/Rise show better statistics than the others, especially below 10 km shown in Figure 12d,e. Thus, the RO event modes did not impact the retrieval quality. The better statistics for GALILEO may be related to the fact that the precision of GALILEO code and phase observations outperforms those of the GPS and GLONASS ones [39,40]. Figure 12a–c show the statistical results obtained for Spire, revealing that below 10 km, the mean fractional differences for GPS/Set, GPS/Rise, GLONASS/Set, and GLONASS/Rise were negative within the magnitude of ~0.8% and maximal STD of ~2.0%, while the mean fractional differences for GALILEO/Set and GALILEO/Rise were negative within the magnitude of ~0.6% and maximal STD of ~1.8%; at heights between 10 and 25 km, the mean fractional differences were negative within the magnitude of ~0.20% and maximal STD of ~0.4%; and above 25 km, the mean fractional differences were positive within the magnitude of ~0.25% and maximal STD of ~0.75%. Within the latitude range of ±45°, a smaller proportion of Spire GNSS RO data were analyzed compared to the global data as shown in Figure 12d,e. Below 10 km, the mean fractional differences for GPS/Set, GPS/Rise, GLONASS/Set, and GLONASS/Rise were negative within the maximum value of ~1.2% and maximal STD of ~2.2%, while the mean fractional differences for GALILEO/Set and GALILEO/Rise were negative within the maximum value of ~0.5% and maximal STD of ~1.8%; at heights between 10 and 25 km, the mean fractional differences were within the magnitude of 0.2%, with a maximal STD of ~0.5%; and above 25 km, the mean fractional differences were positive within the maximum value of ~0.3% and maximal STD of ~0.8%. For the COSMIC-2 data shown in Figure 12g–i, below 10 km, the mean fractional differences were negative within the maximum value of ~1.8% and maximal STD of ~2.0%; at heights between 10 and 25 km, the mean fractional differences were within the magnitude of ~0.2% and maximal STD of ~0.7%; and above 25 km, the mean fractional differences were positive within the maximum value of ~0.2% and maximal STD of ~1.0%.

It is evident that the Spire data collected within latitudes of ±45° has comparable quality to the COSMIC-2 data. Beyond the ±45° latitude range, the retrieval quality of the Spire data was higher than that of the COSMIC-2 data, perhaps because less moisture is present at higher latitudes.

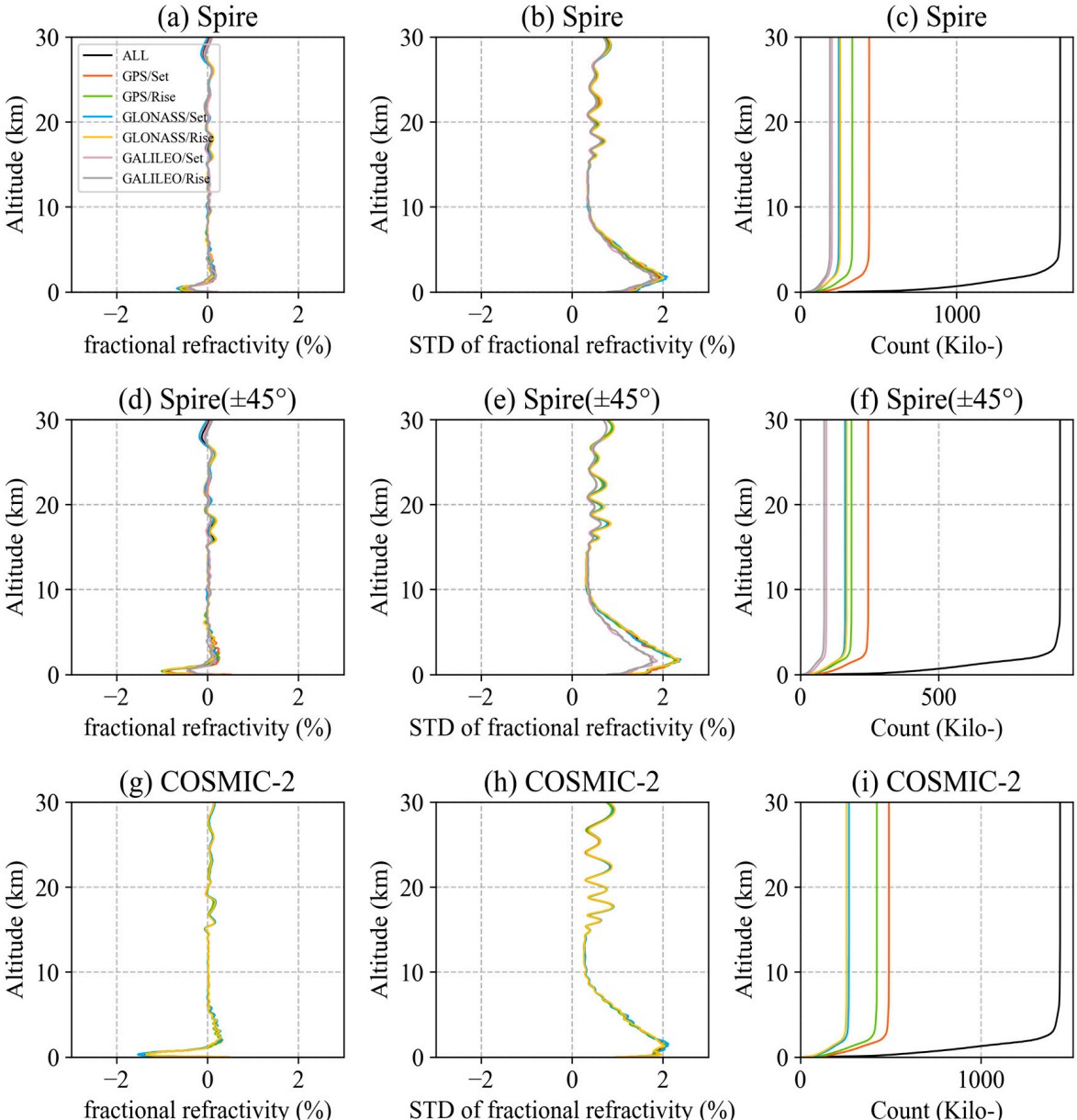

**Figure 12.** Neutral atmospheric refractivity comparison of the Spire and COSMIC-2 wetPf2 products with ERA5 data from DOY 351 in 2020 to 015 in 2021 below 30 km (black: all data, red: GPS/Set, green: GPS/Rise, blue: GLONASS/Set, yellow: GLONASS/Rise, purple: GALILEO/Set, and gray: GALILEO/Rise). (**a–c**) Mean fractional differences, STDs, and counts in the Spire record, (**d–f**) mean fractional differences, STDs, and counts in the Spire record (±45°), and (**g–i**) mean fractional differences, STDs, and counts in the COSMIC-2 record.

Furthermore, we performed a statistical comparison of the specific humidity, absolute temperature, and pressure obtained from RO events through the 1DAR method with those of the ERA5 dataset under the same circumstances. Figure 13 shows the mean differences and STDs in the meteorological parameters retrieved from the Spire (±45°) and COSMIC-2 data. The mean differences and STDs of both the Spire and COSMIC-2 data exhibit identical statistical results, especially above the 10 km altitude. Below 10 km, the mean differences in specific humidity for both the Spire and COSMIC-2 data were negative, with maximal values of ~0.45 g/kg and ~0.70 g/kg and maximal STDs of ~0.9 g/kg and ~0.8 g/kg, respectively; the mean differences in temperature in the Spire and COSMIC-2 data were positive within ~0.25 K and ~0.4 K with maximal STDs of ~0.7 K and ~0.7 K,

respectively, and the mean differences in pressure in the Spire and COSMIC-2 data were negative within ~0.2 hPa and ~0.1 hPa, with maximal STDs of ~1.1 hPa and ~1.3 hPa, respectively. Above 10 km, according to the statistics obtained for the Spire and COSMIC-2 data, the specific humidity exhibited mean differences and STDs of nearly equal zero; the temperature exhibited mean differences and STDs that fluctuated near zero within a maximum value of ~0.3 K and a maximal STD of ~1.8 K, and the pressure exhibited mean differences and positive STDs within the maximum value of ~0.1 hPa and maximal STD of ~0.5 hPa, indicating very similar results between the two data sources. Compared to the ERA5 product, the specific humidity, temperature, and pressure of Spire (±45°) and COSMIC-2 indicated identical retrieval qualities.

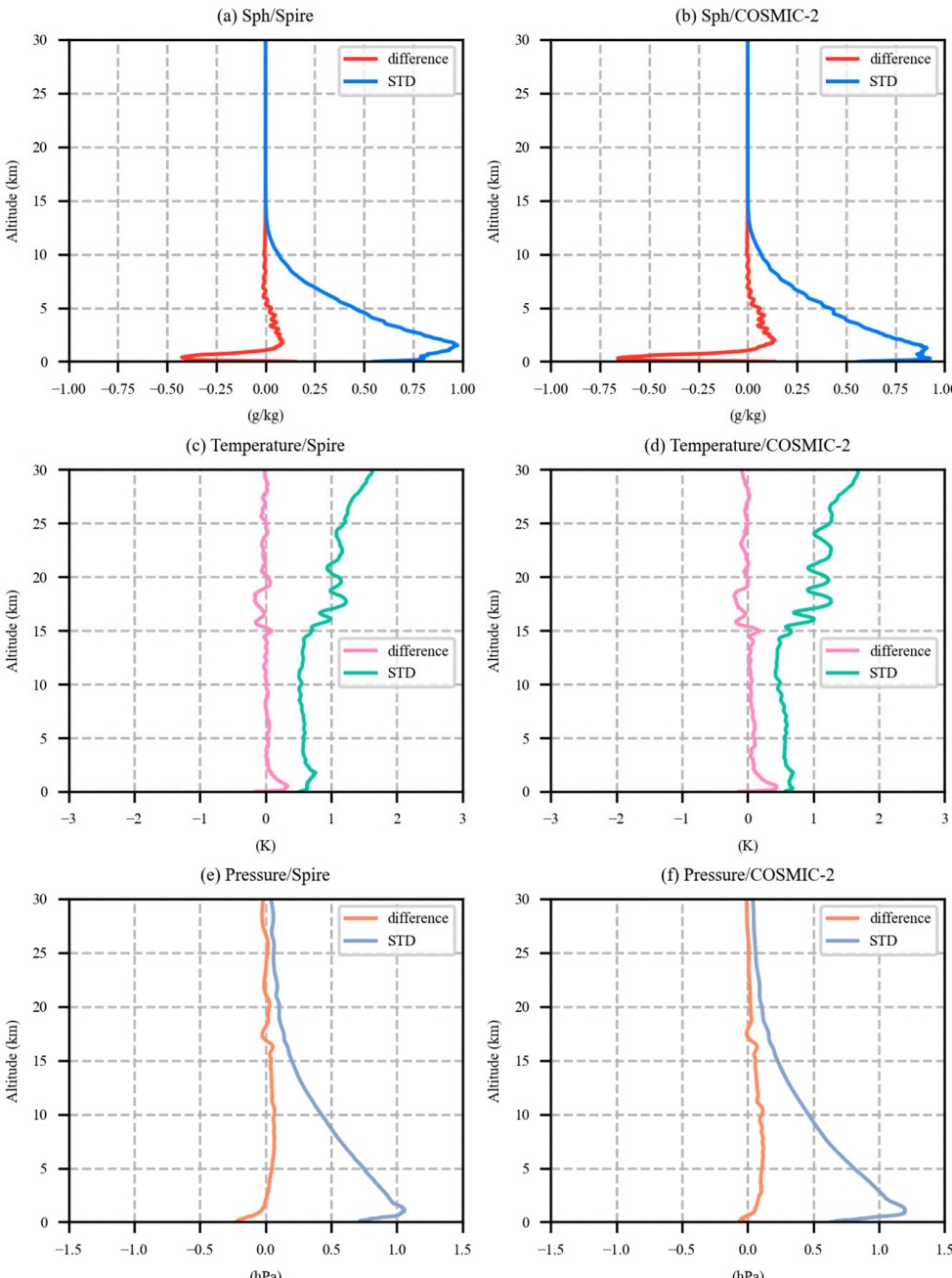

**Figure 13.** Meteorological parameter comparison of the Spire (±45°) and COSMIC-2 wetPf2 products compared with ERA5 data from DOY 351 in 2020 to 015 in 2021 below 30 km ((**a**,**b**) specific humidity for Spire and COSMIC-2, respectively; (**c**,**d**) absolute temperature for Spire and COSMIC-2, respectively; (**e**,**f**) pressure for Spire and COSMIC-2, respectively).

To remove the coupling effect that arises when assimilating data into the ERA5 dataset, Spire (±45°) and COSMIC-2 data were compared with radiosonde data. Above an altitude of approximately 25 km, insufficient radiosonde data were used to perform the statistical comparison; thus, at this height, the results were not credible. The mean differences in the refractivity, specific humidity, temperature, and pressure of the Spire data were negative within ~1.8%, ~0.80 g/kg, ~0.45 K, and ~1.0 hPa below 10 km, respectively, with maximal STDs of ~3.0%, ~1.8 g/kg, ~1.7 K, and ~2.2 hPa, respectively, as shown in Figure 14. Above the altitude of 10 km, the mean differences in refractivity, specific humidity, temperature, and pressure in the Spire data fluctuated near zero within 0.2%, 0.1 g/kg, ~0.2 K, and ~0.3 hPa, respectively, with maximal STDs of ~1.0%, ~0.1 g/kg, ~1.3 K, and ~0.7 hPa, respectively.

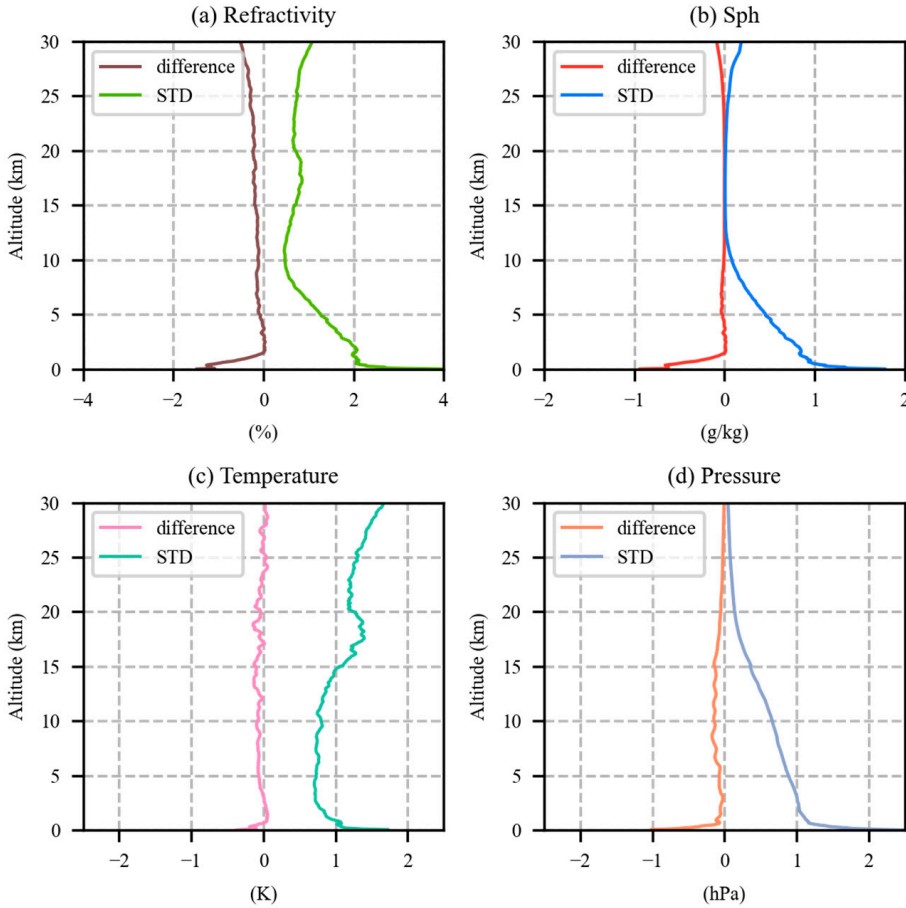

**Figure 14.** Mean differences and STDs of the Spire (±45°) wetPf2 product compared to radiosonde data from DOY 060 in 2022 to 059 in 2023 below 30 km: (**a**) refractivity, (**b**) specific humidity, (**c**) absolute temperature, and (**d**) pressure.

Figure 15 shows that the mean differences in refractivity, specific humidity, temperature, and pressure in the COSMIC-2 data were negative within ~2.2%, ~1.2 g/kg, ~0.2 K, and ~0.2 hPa below 10 km, with maximal STDs of ~2.8%, ~1.5 g/kg, ~1.7 K, and ~1.2 hPa, respectively. Above 10 km, the mean differences in refractivity, specific humidity, and pressure in the COSMIC-2 data fluctuated near zero within values of 0.1%, 0.1 g/kg, and ~0.1 hPa, with maximal STDs of ~0.3%, ~0.2 g/kg, and ~0.8 hPa below 10 km, respectively. The difference in temperature was negative above 10 km within ~0.3 K and with a maximal STD of ~1.3 K. Apparently, compared to the radiosonde data, the retrieval quality of the Spire and COSMIC-2 data exhibited similar statistics, and in the refractivity comparison, the COSMIC-2 data had slightly smaller differences than the Spire data.

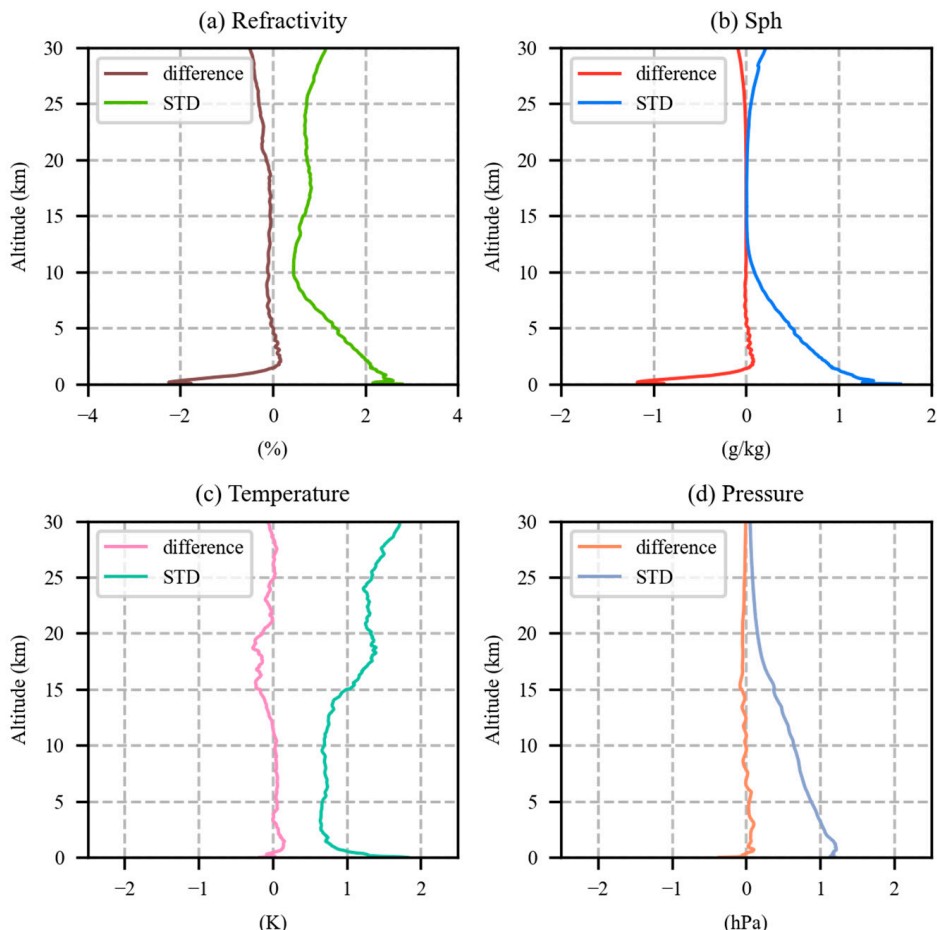

**Figure 15.** Mean differences and STDs of the COSMIC-2 wetPf2 product compared with radiosonde data from DOY 060 in 2022 to 059 in 2023 below 30 km: (**a**) refractivity, (**b**) specific humidity, (**c**) absolute temperature, and (**d**) pressure.

## 4. Discussion

The quality of Spire and COSMIC-2 RO data is assessed Radio Occultation (RO) data is assessed comprehensively.

Nowadays, Spire can receive three navigation systems: GPS, GLONASS, and GALILEO, while the COSMIC-2 is limited to GPS and GLONASS. The initial analysis of GNSS-RO data from Spire and COSMIC-2 shows that Spire can offer a broader global coverage of RO events because of consisting of a diverse set of orbits. GPS-derived RO events were more than GLONASS and GALILEO, due to the greater number of operational GPS satellites. Although there is no significant difference in the number of satellites in orbit for GLONASS and GALILEO, GLONASS-derived RO events for Spire slightly outnumber those derived from GALILEO due to some Spire satellites, such as S128, S115, S117, and others, not receiving GALILEO signals.

The results of SNR indicate that the ability of the Spire to track only GPS signals is significantly weaker than that of the COSMIC-1, and with the joint consideration of GLONASS and GALILEO, Spire can achieve a slightly weaker capability than COSMIC-1. The COSMIC-2 has a superior ability to track signals compared to both the COSMIC-1 and Spire.

COSMIC-2 outperforms Spire in achieving better penetration, primarily attributed to COSMIC-2's higher SNR. Moreover, setting occultation events consistently showed greater penetration depths than rising occultation events for both Spire and COSMIC-2, regardless of the satellite navigation system used. After fixation on the influence of topography, such as mountains, on penetration depth, the Spire and COSMIC-2 data below an altitude of

1 km make up 88.7% and 85.3% of all the data, respectively. Due to topographic changes and water vapor variations with increasing latitude, the penetration depth is affected, thus leading to the retrieval statistics. Through comparing the fixed Spire data within the lower-latitude range of ±45° to the fixed COSMIC-2 data, it is found that the fixed Spire (±45°) data below the 1 km altitude accounted for 84.2% of all the data. In ascending sequence, the ability of the systems to perform deeper soundings could be ranked as follows: Spire fixed (±45°), COSMIC-2 fixed, and Spire fixed. Therefore, COSMIC-2 has a better performance in sounding the deeper troposphere than Spire.

With the ERA5 and Radiosonde as the reference data, it is evident that the Spire data collected within latitudes of ±45° has comparable quality to the COSMIC-2 data. Beyond the ±45° latitude range, the retrieval quality of the Spire data was higher than that of the COSMIC-2 data, perhaps because less moisture is present at higher latitudes. Additionally, the analysis of mean and STD of fractional differences in refractivity across different seasons does not reveal significant seasonal variations in refractivity errors. In addition, Spire can produce a great number of atmospheric profiles with quality comparable to that of COSMIC-2 to complement the limitation of COSMIC-2 and cover the low-latitude area (±45°).

## 5. Conclusions

In this study, we mutually analyzed the coverage, SNR, and penetration characteristics of Spire and COSMIC-2 data and assessed the corresponding RO neutral-atmospheric products through comparisons with ERA5 and radiosonde datasets considering the division of GNSS and RO modes. Based on the above assessment and analysis, the conclusions are as follows:

- Spire's RO events demonstrated global coverage due to various orbiting geometries, while COSMIC-2 events were concentrated in the tropics and reduced at higher latitudes.
- GPS-derived RO events were generally more abundant than GLONASS-derived events in both Spire and COSMIC-2 datasets. And GLONASS-derived RO events slightly outnumbered those derived from GALILEO for Spire.
- STRATOS payload on Spire, equipped with lower-gain antennas, exhibited weaker signal capturing compared to IGOR (COSMIC-1) and significantly weaker than TRGS (COSMIC-2).
- The SNR averages of the GLONASS-derived RO events in the Spire data are much stronger than those of the GPS-derived events, while for COSMIC-2, the strengths of the SNR averages had the same magnitudes, with little difference observed between the GPS- and GLONASS-derived RO events.
- In the same coverage area (±45°), COSMIC-2 demonstrated better penetration ability than Spire.
- Based on the research by Gorbunov et al. (2022) [27], it has been revealed that the SNR serves as an indicator of signal strength and holds a crucial role in penetration. Penetration depth was found to be influenced by SNR, GNSS, RO modes, topography, and latitude, as revealed by combined results obtained in Sections 3.2 and 3.3.
- Compared to the ERA5 and radiosonde products, the Spire and COSMIC-2 datasets have identical retrieval qualities when considering the RO data of Spire and COSMIC-2. The accuracy of the neutral-atmosphere Spire data products acquired herein was comparable with those of COSMIC-2.

Corroborated the COSMIC-2 retrieval quality assessments made by Ho et al. (2020), our study contributes to the understanding of the capabilities and performance of Spire and COSMIC-2 RO retrievals. These findings emphasize the valuable role of nanosatellite GNSS-RO techniques like Spire in advancing atmospheric monitoring. Incorporating commercial initiatives such as Spire supplements scientific GNSS-RO data and addresses the need for global observing systems.

**Author Contributions:** Conceptualization, C.Q.; methodology, C.Q.; software, C.Q.; validation, C.Q., K.Z., and J.Z.; formal analysis, Y.C.; investigation, H.L.; resources, X.W.; data curation, D.L.; writing—original draft preparation, C.Q.; writing—review and editing, X.W.; supervision, H.Y.; project administration, X.W.; funding acquisition, X.W. All authors have read and agreed to the published version of the manuscript.

**Funding:** This research was supported by the funding program from the Aerospace Information Research Institute and CAS Pioneer Hundred Talents Program.

**Data Availability Statement:** The Spire and COSMIC-2 RO data involved in this study are available to freely download from the CDAAC at https://www.cosmic.ucar.edu/ accessed on 21 October 2023. ERA5 data can be downloaded publicly from the provided URL https://cds.climate.copernicus.eu/cdsapp#!/dataset/reanalysis-era5-pressure-levels?tab=form accessed on 21 October 2023. The IGRA2 data are accessible at the website https://www.ncei.noaa.gov/pub/data/igra/ accessed on 21 October 2023.

**Conflicts of Interest:** The authors declare no conflict of interest.

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
