# Peer review of "Comparative Assessment of Spire and COSMIC-2 Radio Occultation Data Quality"

_remotesensing, doi:10.3390/rs15215082_

Round 1

Reviewer 1 Report (New Reviewer)

This manuscript presents a very comprehensive study of the space-borne GNSS RO data quality for two major constellations, one from COSMIC-2 and one from the cubic-satellite Spire. The study is well-organized and delivers valuable information about the GNSS constellation, RO type, SNR, penetration depth, and data quality. It covers pretty much all aspects of the GNSS RO neutral atmospheric product quality. There are 17 figures and 4 tables in total, which is a lot. In general, I think this manuscript is publishable. However, the authors should adjust the content of the manuscript to make it more concentrated on one or a few topics rather than everything. 

  First and most important issue: I think only section 3.5 matches what the title described. Almost everything before section 3.5 has nothing to do with ERA5 and radiosonde. Please either revise the title or adjust the sections.    Many sections, such as 3.1, 3.2, and 3.3, can be more concise. Practically, we do not care much about the differences among different satellites, especially for Spire, as they won’t last too long. Figure 8 shows very similar patterns among different satellites. The figures can be summarized as a few numbers for the whole constellation. You could think from the perspective of the data users.   I think the content in section 3.5 covers the most interesting topics. I have specific comments on those figures.    Figure 12, why the refractivity is dimensionless or unitless. It has the unit of the typical N-unit, you should create a percentage difference as the refractivity is almost exponential. There is little to zero difference above 10 km, as the refractivity is small around that altitude. Percentage or fractional difference could show the difference, if any. Also, the mean and STD should be included.    Figure 13: make it clear the RMS is for the absolute difference or percentage/fractional difference.   Figure 14: such comparison statistics between RO and ERA5 have been shown in many studies, I never saw many wary and waggle features like this.  For 10K+ profiles, the statistics should be much smoother. There seems to be something wrong in the comparison, such as the spikes around 17.5 km showing up in both datasets (COSMIC-2 and Spire). I suppose there is something wrong with the ERA5.   Figure 15: same as above, what are those waggles in the differences, about 15 km for temperature and all altitudes for pressure?    Figures 16 and 17: how the RO vs. radiosonde pair are determined for the comparison and how many pairs are retrieved. I expect much smoother results in the difference if they are the statistics of many pairs. 

Author Response

Please refer to the attachment

Reviewer 2 Report (New Reviewer)

The authors compared the Spire and COSMIC-2 Radio Occultation (GPS and GLONASS) Neutral Atmospheric Products to analyze the coverage, SNR, and penetration characteristics. Although Spire's RO events show global coverage because of various orbiting geometries, COSMIC-2 events are concentrated in the tropics based on the target area of the system. The payload in the Spire is significantly weaker than that of COSMIC-2. Moreover, COSMIC-2 demonstrated better penetration ability than Spire. From the combined results, the authors infer that penetration depth to be influenced by SNR, GNSS, RO modes, topography, and latitude. The accuracy of neutral data products from Spire and COSMIC-2 agrees well with the ERA5 and radiosonde products.

The results found in this study are reasonable but with very limited (less than a month) datasets. It could have covered full seasons of datasets to the capability in different seasons.

The authors should clarify how the commercial data (Spire) would complement the efforts made by the lead COSMIC-2 program for global coverage of lower and upper atmospheric products.

A selection of the combination of red and purple combinations for respective GPS and GLONASS occultation events is not a good choice to distinguish between the two even if the point types are different.

Line 221 A space is expected before and in “68%and 32%, respectively”, Similarly in Line 282 a space is expected after pronounced in “…disturbances are more pronounced[21, 27].” Please correct all such occurrences throughout the manuscript.

In Fig. 9c and d, the authors need to justify why the dense scattered is observed above the Indian landmass. Similarly in Fig 10 also, please justify.

Did the authors use COSMIC-1 data also? Please clarify.

Line 460 Correct “(b) counts (Kilo-).”

 I would recommend a minor revision for further improvement of this manuscript by incorporating the comments and suggestions.

Round 2

Reviewer 1 Report (New Reviewer)

I have no more questions or concerns after those RO vs. ERA5 comparison statistics figures are fixed.

This manuscript is a resubmission of an earlier submission. The following is a list of the peer review reports and author responses from that submission.

Round 1

Reviewer 1 Report

Comments to authors:

This study comprehensively analyzed the different characteristics, such as the coverage, SNR, and penetration, of Spire and COSMIC-2 products. Additionally, it also assessed the corresponding radio occultation neutral-atmospheric products by using the ERA5 and radiosonde datasets as external references. Overall, the research effort invested in this endeavor has been substantial. Nonetheless, prior to recommending its publication in the journal, some moderate modifications should be carefully conducted to further improve the quality of this manuscript. The specific comments are listed as follows:

1. Abstract:

1)     Some abbreviations defined in the Abstract are not necessarily required, as they have not been used in this part, e.g., ECMWF.

2)     Line 16: The acronyms of GNSS and RO should not be simultaneously defined.

3)     Line 21: The use of “GNSSs” is NOT normal practice, as GNSS generally refers to Global Navigation Satellite Systems.

4)     Line 21: I would suggest adding the word “factors” to summarize all the details you listed.

5)     Line 23: the expression of “reanalysis field” is not widely accepted.

6)     Line 24: I would suggest replacing the word “precision” with “accuracy”, as they have different definitions.

2. In general, it is obviously NOT a good practice to use SO MANY abbreviations in the manuscript. To improve the readability of this manuscript and keep it reader friendly, I would strongly suggest removing most of the abbreviations used in the current version. Only abbreviations that are widely accepted among readers (from different research directions and even disciplines) can be used. I have roughly counted the number of abbreviations you used in this manuscript, almost 3/4 of them are not necessary. From another aspect, although you have defined many abbreviations throughout the whole manuscript, most of them are not used in the following context. Please go through the whole manuscript and carefully revise them.

3. Line 163: The expression of “DOY xxxx.xxx” would result in confusion. Please change these expressions or explain them further. Why not simply use “over the xx-day study period” (see also Lines 97-98)?

4. Figure 1/2. Please further illustrate the definitions of the different signs shown in these figures. For example, “red dots represent xxx; purple triangles denote xxx.”

5. Figures 6/7: there are some overlaps in these figures. Please replot them.

6. Figure 10: Please clearly define the content of this figure. The title is not acceptable.

.

Reviewer 2 Report

This study analyzed the coverage, SNR, and penetration characteristics of Spire and COSMIC-2 data and assessed the corresponding RO neutral-atmospheric products through comparisons with ERA5 and radiosonde datasets. The paper made a contribution to show that the precision of Spire's neutral-atmosphere data products was found to be comparable to that of COSMIC-2. This reviewer has only one minor comment.

1.    In the data and methodology part, the authors are suggested to add some brief descriptions on the SPIRE and COSMIC-2 missions.

Reviewer 3 Report

This study presents a well-designed comparison between Spire and COSMIC-2 products using ERA5 and radiosonde data as references. Nevertheless, a few minor improvements could enhance its clarity and completeness.

Figure Captions: Figure 3 and Figure 4 captions need slight refinement. In Figure 3, clarify the x-axis labels for the two kinds of orbits in the Spire dataset: "37DO x-axis labels are depicted in blue, and SSO x-axis labels are represented in black." This adjustment ensures clarity in the caption.

Occultation Events Description: The significance of describing occultation events detected by GPS and GLONASS satellites in chapter 3.1 should be addressed. Currently, the rationale behind this description is unclear. While a explanation is reserved for a later chapter, a brief mention here would add clarity to its necessity.

Uniform X-axis Scales: To facilitate effective comparison, consider making the x-axis scales for figures 16 and 17 a) b) c) and d) identical, respectively. This adjustment enhances the visual comparability of the data presented.

Inclusion of Prior Study Results in Conclusions: The conclusions section could benefit from referencing results from other studies mentioned in the Introduction. Incorporating these findings further strengthens the overall conclusion and highlights the study's contribution within the broader context.

These revisions enhance the study's cohesiveness and ensure that readers grasp the significance of the research findings.

Reviewer 4 Report

The manuscript analysed the comparison of GNSS-RO, COSMIC-2, and ERA-5 datasets globally. The work analyses the datasets and compare their performance with great details. I have following suggestions for the further improvements of the work:

1.      In the data section, kindly mention the period of data being analysed for the present work.

2.      Can you introduce the L1 and L2 for Figure 5 and Set/Rise for Figure 6in the text. All the readers may not be familiar and understanding the explanations with knowing what the significance of L1, L2 is and Rise/set for the work.

3.       The results are unbelievably close for ERA5 and COSMIC2, in Figure 15, and subsequently for Fig. 16 and others in next subsections. Did authors calculated mean profiles for comparison with some data quality controls implemented? If so, kindly mention the conditions used for such quality checks.

4.      Conclusions need to be revised and elaborate. If possible, make them in bullet points.

5.      Also, kindly highlight How the article is approaching the comparisons in a new way (if it is so) and how the article will add new knowledge to the existing one in the final para.
